# Scalable Deep Compressive Sensing

**Zhonghao Zhang**                                                    *zhonghaozhang@yeah.net*
*School of Information and Communication Engineering, University of Electronic Science and Technology of China (UESTC), Chengdu, China*

**Yipeng Liu** *                                                     *yipengliu@uestc.edu.cn*
*School of Information and Communication Engineering, University of Electronic Science and Technology of China (UESTC), Chengdu, China*

**Xingyu Cao**                                                  *shutong.cxy@alibaba-inc.com*
*Alibaba DAMO Academy*

**Fei Wen**                                                           *wenfei@sjtu.edu.cn*
*Department of Electronic Engineering, Shanghai Jiao Tong University, Shanghai, China*

**Ce Zhu**                                                            *eczhu@uestc.edu.cn*
*School of Information and Communication Engineering, University of Electronic Science and Technology of China (UESTC), Chengdu, China*

**Reviewed on OpenReview:** *https://openreview.net/forum?id=10JdgrzNOk*

## Abstract

Deep learning has been used to image compressive sensing (CS) for enhanced reconstruction performance. However, most existing deep learning methods train different models for different subsampling ratios, which brings an additional hardware burden. In this paper, we develop a general framework named scalable deep compressive sensing (SDCS) for the scalable sampling and reconstruction (SSR) of all existing end-to-end-trained models. In the proposed way, images are measured and initialized linearly. Two sampling matrix masks are introduced to flexibly control the subsampling ratios used in sampling and reconstruction, respectively. To achieve a reconstruction model with flexible subsampling ratios, a training strategy dubbed scalable training is developed. In scalable training, the model is trained with the sampling matrix and the initialization matrix at various subsampling ratios by integrating different sampling matrix masks. Experimental results show that models with SDCS can achieve SSR without changing their structure while maintaining good performance, and SDCS outperforms other SSR methods.

## 1 Introduction

Compressive sensing (CS) is a technique that simultaneously samples and compresses signals. And the signal is sampled and reconstructed at a ratio that can be much lower than the Nyquist rate. The sampling process of CS can be expressed as $\mathbf{y} = \mathbf{A}\mathbf{x}$ , where $\mathbf{x} \in \mathbb{R}^N$ is the original signal, $\mathbf{y} \in \mathbb{R}^M$ denotes the measurement, $\mathbf{A} \in \mathbb{R}^{M \times N}$ is the sampling matrix with $M < N$ and $M/N$ is the CS ratio. The signal recovery from $\mathbf{y}$ is under-determined, and it is usually carried out by solving an optimization problem as follows:

$$\min_{\mathbf{x}} \mathfrak{R}(\mathbf{x}), \text{ s. t. } \mathbf{y} = \mathbf{A}\mathbf{x}, \tag{1}$$

where $\mathfrak{R}(\mathbf{x})$ is the regularization term. In this paper, we mainly focus on the visual image CS (Lohit et al., 2018a) which has been applied in single-pixel imaging (SPI) (Lohit et al., 2018a; Duarte et al., 2008) and

---

*Corresponding author

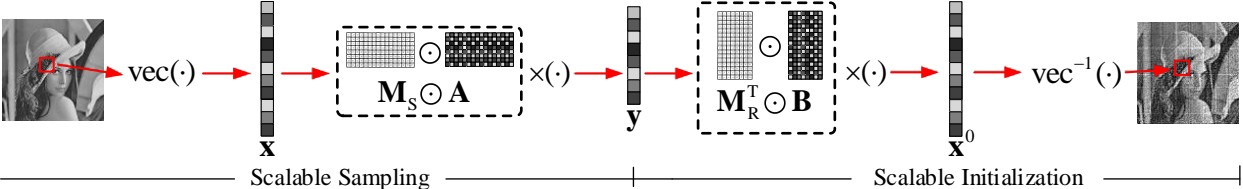

Figure 1: Scalable sampling and scalable initialization of an image block.

wireless broadcast (Yin et al., 2016; Li et al., 2013; Guo et al., 2020). And since block-by-block sampling and reconstruction (Dong et al., 2014; Dinh & Jeon, 2017; Lohit et al., 2018a; Zhang & Ghanem, 2018; Zhang et al., 2021) would bring less burden to the hardware, we mainly focus on the block-based visual image CS problem.

To solve the problem (1), model-based methods (Dong et al., 2014; Li et al., 2020) introduce various hand-crafted regularizers (Elad, 2010; Liu et al., 2019) and apply non-linear iterative algorithms (Beck & Teboulle, 2009; Donoho et al., 2009) to recover images. These methods usually have theoretical guarantees and work well using sampling matrices with different CS ratios. However, their performance needs to be further improved.

In recent years, deep learning has achieved great success in computer vision (Rick Chang et al., 2017; Dong et al., 2018; Yan et al., 2020a;c;b; Tu et al., 2022; Hu et al., 2021). Among them, models for visual image CS can be cast into two categories: traditional deep learning models and deep unfolding models. Traditional deep learning models (Mousavi et al., 2015; Lohit et al., 2018a; Shi et al., 2019a; 2020) are usually stacked by non-linear computational layers. These models map the measurement to the output without considering the prior information of images. Although they can reconstruct high-quality images at a high speed, there is no good interpretability and theoretical guarantee (Huang et al., 2018). Deep unfolding models denote a series of models constructed by mapping iterative algorithms with unfixed numbers of steps onto deep neural networks with fixed numbers of steps (Gregor & LeCun, 2010; Zhang & Ghanem, 2018; Metzler et al., 2017; Zhang et al., 2021; Dong et al., 2018; Sun et al., 2016; Ma et al., 2019). By combining the interpretability of model-based methods and the trainable characteristics of traditional deep learning models, they make a good balance between reconstruction performance and interpretability.

Usually, the above two kinds of deep-learning-based models are trained end-to-end using some well-known backpropagation algorithms (Kingma & Ba, 2015). However, a common shortage of most existing end-to-end-trained models is that different models have to be trained for different CS ratios. In some applications, sampling and reconstructing images at different CS ratios may be required(Yin et al., 2016; Li et al., 2013). However, storing more than one model with the same structure would bring additional burdens to the hardware. Thus, sampling and reconstructing images at different CS ratios with only one model is needed.

At present, there exist a few methods (Xu et al., 2020; Su & Lian, 2020; Shi et al., 2019b; Zhang et al., 2020; Lohit et al., 2018b; Li et al., 2020) which reconstruct images at different CS ratios using only one model, and they can be roughly cast into two categories. The first kind (Xu et al., 2020; Su & Lian, 2020) trains a single model to adapt to a set of sampling matrices with different CS ratios. The second kind (Shi et al., 2019b; Lohit et al., 2018b) applies only one sampling matrix and integrates its rows to achieve sampling and reconstruction at different CS ratios, and we call such a strategy as **s**calable **s**ampling and **r**econstruction (SSR) in this paper. This paper focuses on SSR methods, because they are more practical in existing applications such as SPI (Lohit et al., 2018a), wireless broadcasting (Yin et al., 2016), and MRI (Sun et al., 2016). In detail, in image CS for wireless broadcast (Yin et al., 2016; Li et al., 2013), images are reconstructed with different quality according to different channel conditions using one sampling matrix at the receiving end. And in SPI (Lohit et al., 2018a), different CS ratios can be applied for different image quality and sampling time by combining rows of the sampling matrix. However, existing SSR methods cannot be applied universally (Shi et al., 2019b) or a more appropriate sampling matrix is needed (Zhang et al., 2020; Lohit et al., 2018b). Therefore, a general and more effective SSR method is expected.

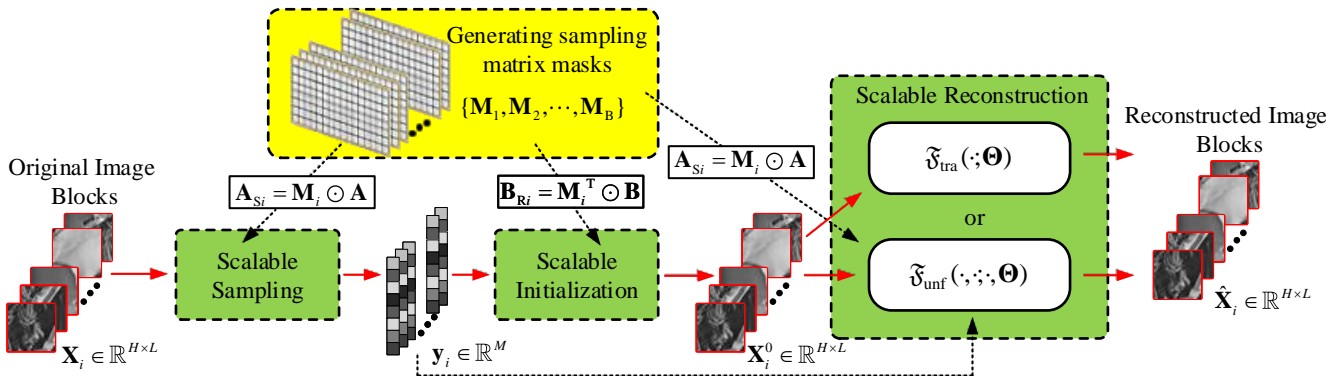

Figure 2: Forward-propogation of the scalable training.

In this paper, we propose a general framework dubbed **s**calable **d**eep **c**ompressive **s**ensing (SDCS) to achieve sampling and reconstructing images at all CS ratios in a certain range. In detail, two binary sampling matrix masks are developed to activate rows of the sampling matrix and the initialization matrix to control the CS ratios for SSR. And we develop a novel training strategy named scalable training, which integrates the multi-ratio information into the training stage by randomly generating sampling matrix masks for different CS ratios. We emphasize that SDCS can bring to the model the ability of SSR, while maintaining the characteristics of its own structure. Furthermore, experimental results show that the model with SDCS can obtain a more effective combination of sampling matrix and model than existing SSR methods.

Our paper has three contributions:

- We propose a framework named SDCS that jointly trains the sampling matrix and the model to achieve sampling and reconstruction at all CS ratios in a certain range.

- With SDCS, a deep learning model can achieve SSR without changing its original structure, while maintaining good performance.

- Technically, SDCS can be used for all end-to-end-trained deep learning models.

## 2 SDCS

In this section, we introduce the proposed framework SDCS which is simple but powerful. SDCS is composed of four parts: scalable sampling, scalable initialization, scalable reconstruction, and scalable training.

### 2.1 Scalable Sampling

Assume that the largest CS ratio is $R_{\mathrm{M}}$, then the sampling matrix can be expressed as $\mathbf{A} \in \mathbb{R}^{\lceil R_{\mathrm{M}} N \rceil \times N}$. It can be noticed that the CS ratio is determined by the row number of $\mathbf{A}$. Therefore, to achieve scalable sampling, we design a sampling matrix mask $\mathbf{M}_{\mathrm{S}} \in \mathbb{R}^{\lceil R_{\mathrm{M}} N \rceil \times N}$ to control the activities of the rows of $\mathbf{A}$. $\mathbf{M}_{\mathrm{S}}$ is a zero-one matrix which satisfies $\mathbf{M}_{\mathrm{S}}(1 : \lceil R_{\mathrm{S}} N \rceil , :) = 1$ and $\mathbf{M}_{\mathrm{S}}(\lceil R_{\mathrm{S}} N \rceil + 1 : \lceil R_{\mathrm{M}} N \rceil , :) = 0$, where $R_{\mathrm{S}}$ denotes the CS ratio for sampling. In such a case, we can generate a new sampling matrix as $\mathbf{A}_{\mathrm{S}} = \mathbf{M}_{\mathrm{S}} \odot \mathbf{A}$, where $\odot$ denotes the element-wise product. Since the $\lceil R_{\mathrm{S}} N \rceil + 1$-th row to the $\lceil R_{\mathrm{M}} N \rceil$-th row of $\mathbf{A}_{\mathrm{S}}$ are all filled with 0, we say that the first row to the $\lceil R_{\mathrm{S}} N \rceil$-th row of $\mathbf{A}$ are activated. In detail, if the original image block is $\bar{\mathbf{X}} \in \mathbb{R}^{H \times L}$ satisfying $N = HL$, then the scalable sampling at the CS ratio of $R_{\mathrm{S}}$ can be expressed as:

$$\mathbf{y} = \mathbf{A}_{\mathrm{S}} \operatorname{vec}(\bar{\mathbf{X}}), \tag{2}$$

where $\text{vec}(\cdot)$ is an operator which transforms a matrix to a vector and $\mathbf{y} \in \mathbb{R}^{\lceil R_{\mathrm{M}} N \rceil}$ is the measurement. It can be noticed that $\mathbf{y}(\lceil R_{\mathrm{S}} N \rceil + 1 : \lceil R_{\mathrm{M}} N \rceil) = 0$ and $\mathbf{y}(1 : \lceil R_{\mathrm{S}} N \rceil)$ is the valid measurement for reconstruction. In this way, we can achieve a unified learning mode with different compression rates.

## 2.2 Scalable Initialization

For deep learning methods, the initialized image is important in the following reconstruction. In SDCS, we use a linear operation to initialize the image block.

Based on some deep-learning based models (Shi et al., 2019a; Zhang & Ghanem, 2018; Zhang et al., 2021), an initialization matrix $\mathbf{B} \in \mathbb{R}^{N \times \lceil R_{\mathrm{M}} N \rceil}$ is developed. Similar to (2), a sampling matrix mask $\mathbf{M}_{\mathrm{R}} \in \mathbb{R}^{\lceil R_{\mathrm{M}} N \rceil \times N}$ is proposed to control the activities of the columns of $\mathbf{B}$, where $\mathbf{M}_{\mathrm{R}}(1 : \lceil R_{\mathrm{R}} N \rceil, :) = 1$ and $\mathbf{M}_{\mathrm{R}}(\lceil R_{\mathrm{R}} N \rceil + 1 : \lceil R_{\mathrm{M}} N \rceil, :) = 0$. $R_{\mathrm{R}}$ denotes the CS ratio for initialization and reconstruction, which satisfies $R_{\mathrm{R}} \leq R_{\mathrm{S}}$. In such a case, we can activate the first column to the $\lceil R_{\mathrm{R}} N \rceil$-th column of $\mathbf{B}$ to generate a new initialization matrix as $\mathbf{B}_{\mathrm{R}} = \mathbf{M}_{\mathrm{R}}^{\mathrm{T}} \odot \mathbf{B}$. The detailed scalable initialization at the CS ratio of $R_{\mathrm{R}}$ can be expressed as:

$$\mathbf{X}^0 = \text{vec}^{-1}(\mathbf{B}_{\mathrm{R}} \mathbf{y}), \tag{3}$$

where $\mathbf{X}^0 \in \mathbb{R}^{H \times L}$ denotes the initialized image block and $\text{vec}^{-1}(\cdot)$ is the operator which transforms a vector to matrix.

In some cases, $R_{\mathrm{R}}$ can be lower than $R_{\mathrm{S}}$. For example, in wireless broadcasting (Yin et al., 2016), images are transferred at a high CS ratio and are received at a low CS ratio due to the poor channel condition. Fig. 1 illustrates the scalable sampling and scalable initialization of an image.

## 2.3 Scalable Reconstruction

In this subsection, we describe the scalable reconstruction of two different kinds of deep learning models: traditional deep learning models and deep unfolding models.

The generalized reconstruction process of traditional deep learning models can be expressed as:

$$\hat{\mathbf{X}} = \mathfrak{F}_{\mathrm{tra}}(\mathbf{X}^0; \boldsymbol{\Theta}), \tag{4}$$

where $\hat{\mathbf{X}} \in \mathbb{R}^{H \times L}$ is the reconstructed image block and $\boldsymbol{\Theta}$ contains trainable parameters of the model. In SDCS, $\boldsymbol{\Theta}$ is trained with $\mathbf{A}$ and $\mathbf{B}$ to make sure that $\mathfrak{F}_{\mathrm{tra}}(\cdot; \boldsymbol{\Theta})$ can perform well at all CS ratios.

The reconstruction model of a deep unfolding model is usually composed of $K$ reconstruction modules with the same structure. In each module, the sampling matrix also participates in the image reconstruction. In detail, the generalized reconstruction process of a deep unfolding model can be expressed as:

$$\hat{\mathbf{X}} = \mathfrak{F}_{\mathrm{unf}}(\mathbf{X^0}, \mathbf{y}; \mathbf{A}, \boldsymbol{\Theta}) = \mathfrak{F}_{\mathrm{unf}}^K(\mathbf{X}^{K-1}, \mathbf{y}; \mathbf{A}, \boldsymbol{\Theta}^K), \tag{5}$$

$$\mathbf{X}^k = \mathfrak{F}_{\mathrm{unf}}^k(\mathbf{X}^{k-1}, \mathbf{y}; \mathbf{A}, \boldsymbol{\Theta}^k), \tag{6}$$

where $\mathfrak{F}_{\mathrm{unf}}(\cdot, \cdot; \mathbf{A}, \boldsymbol{\Theta})$ is the entire deep unfolding model, of which $\boldsymbol{\Theta}$ contains its trainable parameters. $\mathfrak{F}_{\mathrm{unf}}^k(\cdot, \cdot; \mathbf{A}, \boldsymbol{\Theta}^k)$ is the $k$-th reconstruction module and $\boldsymbol{\Theta}^k$ contains its trainable parameters. the inputs of $\mathfrak{F}_{\mathrm{unf}}(\cdot, \cdot; \mathbf{A}, \boldsymbol{\Theta})$ and $\mathfrak{F}_{\mathrm{unf}}^k(\cdot, \cdot; \mathbf{A}, \boldsymbol{\Theta}^k)$ usually contain the image block $\mathbf{X}^0$ and the measurement $\mathbf{y}$. Since $\mathbf{A}$ plays an important role in each reconstruction module, the scalable reconstruction of the deep unfolding model is achieved by applying an activated sampling matrix. In detail, the scalable reconstruction of the $k$-th reconstruction model can be expressed as:

$$\mathbf{X}^k = \mathfrak{F}_{\mathrm{unf}}^k(\mathbf{X}^{k-1}, \mathbf{y}; \mathbf{M}_{\mathrm{R}} \odot \mathbf{A}, \boldsymbol{\Theta}^k). \tag{7}$$

Similar to the traditional deep learning models, $\boldsymbol{\Theta} = \{\boldsymbol{\Theta}^1, \boldsymbol{\Theta}^2, \cdots, \boldsymbol{\Theta}^K\}$ is trained with $\mathbf{A}$ and $\mathbf{B}$. Since the sampling matrix $\mathbf{A}$ usually appears in the image sampling and reconstruction of deep unfolding models, deep unfolding models have great potential to achieve SSR.

---

**Algorithm 1** Scalable training of one epoch.

---

**Input:** training set $\mathbb{T}$, batch size $B$, loss function $L$, max CS ratio $R_{\mathrm{M}}$, sampling matrix $\mathbf{A}$, initialization matrix $\mathbf{B}$, reconstruction model $\mathfrak{F}_{\mathrm{tra}}(\cdot;\mathbf{\Theta})$ or $\mathfrak{F}_{\mathrm{unf}}(\cdot;\mathbf{A},\mathbf{\Theta})$.

**Output:** trained parameters.

1: $\mathbb{T}' \leftarrow \emptyset$
2: **repeat**
3:      Select $\mathbb{S} = \{\mathbf{X}_1, \mathbf{X}_2, \cdots, \mathbf{X}_B\} \in \mathbb{T} \setminus \mathbb{T}'$.
4:      $\mathbb{T}' \leftarrow \mathbb{T}' \cup \mathbb{S}$.
5:      Generate $\{R_1, R_2, \cdots, R_B\}$ randomly, where $R_i \in [1, R_{\mathrm{M}}]$.
6:      Generate $\{\mathbf{M}_1, \mathbf{M}_2, \cdots, \mathbf{M}_B\}$, where $\mathbf{M}_i(1:\lceil R_i N \rceil, :) = 1$ and $\mathbf{M}_i(\lceil R_i N \rceil + 1 : \lceil R_{\mathrm{M}} N \rceil, :) = 0$.
7:      Generate $\mathbb{A}_{\mathrm{S}} = \{\mathbf{A}_{\mathrm{S}1}, \mathbf{A}_{\mathrm{S}2}, \cdots, \mathbf{A}_{\mathrm{S}B}\}$, $\mathbb{A}_{\mathrm{R}} = \{\mathbf{A}_{\mathrm{R}1}, \mathbf{A}_{\mathrm{R}2}, \cdots, \mathbf{A}_{\mathrm{R}B}\}$ and $\mathbb{B}_{\mathrm{R}} = \{\mathbf{B}_{\mathrm{R}1}, \mathbf{B}_{\mathrm{R}2}, \cdots, \mathbf{B}_{\mathrm{R}B}\}$,
        where $\mathbf{A}_{\mathrm{S}i} = \mathbf{M}_i \odot \mathbf{A}$, $\mathbf{A}_{\mathrm{R}i} = \mathbf{M}_i \odot \mathbf{A}$ and $\mathbf{B}_{\mathrm{R}i} = \mathbf{M}_i^{\mathrm{T}} \odot \mathbf{B}$.
8:      **for** $i = 1 : B$ **do**
9:          $\mathbf{y}_i = \mathbf{A}_{\mathrm{S}i} \operatorname{vec}(\mathbf{X}_i)$
10:         $\mathbf{X}_i^0 = \operatorname{vec}^{-1}(\mathbf{B}_{\mathrm{R}i}\mathbf{y}_i)$
11:         $\hat{\mathbf{X}}_i = \mathfrak{F}_{\mathrm{tra}}(\mathbf{X}_i^0; \mathbf{\Theta})$ or $\hat{\mathbf{X}}_i = \mathfrak{F}_{\mathrm{unf}}(\mathbf{X}_i^0, \mathbf{y}_i; \mathbf{A}_{\mathrm{R}i}, \mathbf{\Theta})$
12:      Compute loss $L$ using $\{\hat{\mathbf{X}}_1, \hat{\mathbf{X}}_2, \cdots, \hat{\mathbf{X}}_B\}$ and $\mathbb{S}$.
13:      Update $\mathbf{A}$, $\mathbf{B}$ and $\mathbf{\Theta}$.
14: **until** $\mathbb{T} \setminus \mathbb{T}' = \emptyset$
15: **return** $\mathbf{A}$, $\mathbf{B}$, $\mathbf{\Theta}$.

---

## 2.4 Scalable Training

As shown in (2), (3) and (7), $\mathbf{A}$ and $\mathbf{B}$ are important in effective SSR. How to obtain an appropriate combination of $\mathbf{A}$, $\mathbf{B}$ and the reconstruction model is the main issue. To this end, we develop a novel training strategy dubbed scalable training to train $\mathbf{A}$, $\mathbf{B}$ with parameters of the reconstruction model jointly.

In scalable training, it is assumed that all parameters are trained using stochastic-gradient-descent-related algorithms like Adam (Kingma & Ba, 2015). If the batch size for training is $B$ and the loss function is $L$, the training process of $\mathbf{A}$, $\mathbf{B}$ and $\mathbf{\Theta}$ of one epoch can be expressed as Algorithm 1. And Fig. 2 illustrates the forward-propagation of the scalable training. The gradients of $\mathbf{A}$ and $\mathbf{B}$ can be computed as follows:

$$\nabla_{\mathbf{A}} L = \frac{1}{B} \sum_{i=1}^{B} \mathbf{M}_i \odot \nabla_{\mathbf{M}_i \odot \mathbf{A}} L, \tag{8}$$

$$\nabla_{\mathbf{B}} L = \frac{1}{B} \sum_{i=1}^{B} \mathbf{M}_i^{\mathrm{T}} \odot \nabla_{\mathbf{M}_i^{\mathrm{T}} \odot \mathbf{B}} L. \tag{9}$$

It can be noticed that the closer to the top of $\mathbf{A}$ or the left of $\mathbf{B}$, the more gradient information for updating is obtained, which makes using $\mathbf{M}_{\mathrm{S}}$ and $\mathbf{M}_{\mathrm{R}}$ for effective SSR possible. It is emphasized that the form of $L$ is not limited by SDCS, but related to the combined reconstruction model.

Furthermore, to validate the trained model, a CS **r**atio **v**alidation **g**roup (RVG) is applied. Each RVG contains $G$ validation CS ratios as $\{R_1, R_2, \cdots, R_G\}$. At the end of each epoch, for each ratio $R_i$, the average PSNR on the validation set can be obtained. And the model with the best average PSNR on RVG is regarded as the model for test.

We emphasize that SDCS has no restriction on the structure of deep learning models, which means it can be combined with any end-to-end-trained model for SSR. However, the final performance is determined by the structure of the reconstruction model.

# 3 Related Works

In this section, we first introduce some deep-learning-based methods for image CS, then some SSR methods are compared with SDCS.

## 3.1 Deep Learning Models for Image CS

For traditional deep learning models, Mousavi et al. (Mousavi et al., 2015) first designed a fully-connected-layer-based stacked denoising autoencoder (SDA) for visual image CS. Lohit et al. (Lohit et al., 2018a) first proposed a six-layers CNN-based model named ReconNet to reconstruct image blocks from measurements. Shi et al. (Shi et al., 2019a) proposed a deeper CNN model named CSNet which has trainable deblocking operations and integrated residual connection (He et al., 2016) for better performance. Furthermore, there are some other models (Du et al., 2019; Yao et al., 2019; Bora et al., 2017; Sun et al., 2020) for visual image CS, and all these models have one thing in common the models for reconstruction are trained end-to-end.

Deep unfolding models are first developed for the sparse coding problem (Gregor & LeCun, 2010; Chen et al., 2018; Borgerding et al., 2017). And inspired by these models, Zhang et al. (Zhang & Ghanem, 2018) developed a deep unfolding model named ISTA-Net for image CS problem by unfolding iterative shrinkage-thresholding algorithm (ISTA) and learning sparse transformation functions. Metzler et al. (Metzler et al., 2017) and Zhang et al. (Zhang et al., 2021) established deep unfolding models named LDAMP and AMP-Net respectively based on approximate message passing (AMP) algorithm, where LDAMP samples and reconstructs the entire image, and AMP-Net measures and recovers an image block-by-block with general trainable deblocking modules. Dong et al. (Dong et al., 2018) designed a model named DPDNN inspired by the half-quadratic splitting (HQS) algorithm for image inverse problems which can be applied to visual image CS. These deep unfolding models apply the sampling matrix for reconstruction and they can also be trained end-to-end.

Some of the above methods discuss the sampling matrix training strategies, including in traditional deep learning models (Mousavi et al., 2015; Shi et al., 2019a) or in the deep unfolding model (Zhang et al., 2021), and they all train their initialization matrices. Although the trained sampling matrices can improve the reconstruction performance, they are designed for the single CS ratio and the performance would decrease seriously when the CS ratio changes for SSR. However, using SDCS, the model and the trained sampling matrix can perform well in all CS ratios in a certain range.

## 3.2 SSR Methods

As far as we know, there exist several SSR methods (Shi et al., 2019b; Lohit et al., 2018b; Zhang et al., 2020). We introduce and compare them with SDCS in the following paragraphs.

Shi et al. (Shi et al., 2019b) proposed a model dubbed SCSNet. SCSNet trains the sampling matrix with the reconstruction model which is composed of seven independent sub-models with the same structure. Each sub-model adapts with a sub-range of CS ratios to make sure that the whole model can achieve SSR at CS ratios from 1% to 50%. And a greedy algorithm is applied to rearrange the rows of the sampling matrix for better reconstruction. However, SCSNet has two weaknesses: 1) The number of parameters is very large due to the existence of multiple sub-models. 2) Based on SCSNet, the existing deep learning models have to change their structure to achieve scalable reconstruction which would bring more burden to the hardware. However, SDCS needs only one model to achieve SSR and it can be applied to all end-to-end-trained models without changing their structures.

Zhang et al. (Zhang et al., 2020) propose a framework named CRA which applies two reconstruction models, of which the first one is for initializing and completing the measurement, and the second one is for further reconstruction. Compared with SDCS, CRA do not train the sampling matrix, and two reconstruction models would introduce more parameters. Furthermore, we emphasize that CRA is essentially a pluggable method, which can be combined with other SSR methods by applying a non-linear model for initialization and measurement completion. Therefore, the experimental comparison between SDCS and CRA is not the focus of our paper.

Lohit et al. (Lohit et al., 2018b) designed a general framework like SDCS named Rate-Adaptive CS (RACS) which does not need to change the structure of the model, and it has three training stages. In stage 1, the model is trained with the sampling matrix at a single CS ratio of $R_{\mathrm{M}}$. And all parameters of the model are frozen after stage 1. In stage 2, The first $R_{\mathrm{K}}N$ rows of the sampling matrix are optimized, where $R_{\mathrm{K}} < R_{\mathrm{M}}$. In stage 3, the following rows of the sampling matrix are trained one by one. It can be noticed that RACS has an obvious weakness: the model is learned for a specific sampling matrix with CS ratio $R_{\mathrm{M}}$ in stage 1, which means the performance of the model at lower CS ratios can be further improved. Different from RACS, with SDCS, the learned model adapts to a sampling matrix that can change its CS ratios from 1% to $R_{\mathrm{M}}$ using a sampling matrix mask. Our strategy brings the model the potential that performs better for SSR.

## 4 Experimental Results

### 4.1 Experimental settings

In this paper, the model combined with SDCS is named as *model*-SDCS. To evaluate the performance of SDCS, six models are combined with SDCS, namely SDA (Mousavi et al., 2015), ReconNet (Lohit et al., 2018a), CSNet$^+$ (Shi et al., 2019a), ISTA-Net$^+$ (Zhang & Ghanem, 2018), DPDNN (Dong et al., 2018) and AMP-Net (Zhang et al., 2021), which sample and reconstruct images block-by-block with the block size of $33 \times 33$ that makes $N = 1089$. SDA, ReconNet, CSNet$^+$ are traditional deep learning models. ISTA-Net$^+$, DPDNN and AMP-Net are deep unfolding models with 9, 6 and 6 reconstruction modules respectively. In this paper, the activation functions of SDA are changed to the Rectified Linear Unit (ReLU) (Lohit et al., 2018a) for better performance. It is worth noting that in CSNet$^+$ and AMP-Net, trainable deblocking operations are applied. SDA, ReconNet, ISTA-Net$^+$ and DPDNN do not train the sampling matrix in their original matrix, and CSNet$^+$ and AMP-Net have the trainable sampling matrix. Furthermore, the above six models have the same initialization matrix in equation 3 of this paper. Furthermore, since SCSNet (Shi et al., 2019b) and RACS (Lohit et al., 2018b) can achieve SSR like *model*-SDCS, they are compared with SDCS to show the effectiveness of our framework.

All of our experiments are performed on two datasets: BSDS500 (Arbelaez et al., 2010) and Set11 (Lohit et al., 2018a). BSDS500 contains 500 colorful visual images and is composed of a training set (200 images), a validation set (100 images) and a test set (200 images). Set11 (Lohit et al., 2018a) contains 11 grey-scale images. In this paper, BSDS500 is used for training, validation and testing. And Set11 is used for testing. We generate two training sets for models with and without trainable deblocking operations. (a) Training set 1 contains 89600 sub-images sized of $99 \times 99$ which are randomly extracted from the luminance components of images in the training set of BSDS500 (Shi et al., 2019a). (b) Training set 2 contains 195200 sub-images sized of $33 \times 33$ which are randomly extracted from the luminance components of images in the training set of BSDS500 (Zhang & Ghanem, 2018). In this paper, CSNet$^+$, AMP-Net, CSNet$^+$-SDCS, AMP-Net-SDCS and SCSNet are trained on training set 1 due to the existence of trainable deblocking operations. SDA, ReconNet, ISTA-Net$^+$, DPDNN, SDA-SDCS, ReconNet-SDCS, ISTA-Net$^+$-SDCS and DPDNN-SDCS are trained on training set 2. The way to combine these models with SDCS is described in Algorithm 1. And they are trained on the conditions in their original papers. Moreover, we use the validation set of BSDS500 for model choosing and the test set of BSDS500 for testing. In this paper, all sampling matrices are initialized randomly in Gaussian distribution. $R_{\mathrm{M}}$ is 50% and RVG is $\{1\%, 4\%, 10\%, 25\%, 30\%, 40\%, 50\%\}$. All experiments are performed on a computer with an AMD Ryzen7 2700X CPU and an RTX2080Ti GPU.

### 4.2 Comparison with original deep learning methods

In this subsection, we compare SDA, ReconNet, CSNet$^+$, ISTA-Net$^+$, DPDNN and AMP-Net with SDA-SDCS, ReconNet-SDCS, CSNet$^+$-SDCS, ISTA-Net$^+$-SDCS, DPDNN-SDCS and AMP-Net-SDCS.AMP-Net-SDCS* represents the experimental results that sampling matrix **A** and initialization matrix **B** are not involved in training. Table 1 and Table 2 show the average PSNR and SSIM of 12 models tested on Set11 and the testing set of BSDS500 at different CS ratios respectively. We emphasize that there are seven

different models for seven different test CS ratios for the method without SDCS, and a single model is tested at different CS ratios for *model*-SDCS.

Table 1: The results of twelve models tested on Set11 at different CS ratios, where the best is marked in bold.

| Method | 50% | 40% | 30% | 25% | 10% | 4% | 1% |
|---|---|---|---|---|---|---|---|
| | PSNR (dB)/SSIM | | | | | | |
| SDA | 26.43/0.8007 | 25.14/0.7371 | 24.77/0.7191 | 24.77/0.7234 | 23.66/0.6794 | 21.05/0.5720 | 17.69/0.4376 |
| SDA-SDCS | **30.80/0.9038** | **30.63/0.9009** | **29.43/0.8793** | **28.76/0.8636** | **25.58/0.7660** | **22.77/0.6458** | **19.87/0.4829** |
| ReconNet | 32.12/0.9137 | 30.59/0.8928 | 28.72/0.8517 | 28.04/0.8303 | 24.07/0.6958 | 21.00/0.5817 | 17.54/0.4426 |
| ReconNet-SDCS | **34.29/0.9532** | **33.81/0.9242** | **32.42/0.9313** | **31.42/0.9173** | **26.90/0.8225** | **23.57/0.6931** | **20.02/0.5071** |
| CSNet$^+$ | **38.19/0.9739** | **36.15/0.9625** | **33.90/0.9449** | **32.76/0.9322** | 27.76/0.8513 | **24.24/0.7412** | 20.09/0.5334 |
| CSNet$^+$-SDCS | 36.65/0.9645 | 35.48/0.9568 | 33.58/0.9414 | 32.44/0.9295 | **27.85/0.8493** | 23.92/0.7303 | **20.32/0.5394** |
| ISTA-Net$^+$ | **38.08**/0.9680 | **35.93**/0.9537 | **33.66**/0.9330 | **32.27**/0.9167 | 25.93/0.7840 | 21.14/0.5947 | 17.48/0.4403 |
| ISTA-Net$^+$-SDCS | 36.51/**0.9693** | 34.92/**0.9587** | 32.85/**0.9400** | 31.65/**0.9256** | **26.99/0.8334** | **23.57/0.7073** | **20.13/0.5146** |
| DPDNN | 35.85/0.9532 | 34.30/0.9411 | 32.06/0.9145 | 30.63/0.8924 | 24.53/0.7392 | 21.11/0.6029 | 17.59/0.4459 |
| DPDNN-SDCS | **39.50/0.9775** | **37.61/0.9686** | **35.38/0.9543** | **34.12/0.9434** | **29.07/0.8708** | **25.08/0.7622** | **20.55/0.5423** |
| AMP-Net | **40.27/0.9804** | **38.23/0.9713** | **35.90**/0.9574 | 34.59/**0.9477** | 29.45/0.8787 | 25.16/0.7692 | **20.57/0.5639** |
| AMP-Net-SDCS* | 34.57/0.9427 | 32.89/0.9249 | 30.12/0.8922 | 29.32/0.8688 | 24.99/0.7201 | 21.21/0.5649 | 18.97/0.4561 |
| AMP-Net-SDCS | 39.67/0.9781 | 37.96/0.9703 | 35.89/**0.9576** | **34.67/0.9477** | **29.59/0.8792** | **25.43/0.7750** | 20.47/0.5629 |

Table 2: The results of twelve models tested on the test set of BSDS500 at different CS ratios, where the best is marked in bold.

| Method | 50% | 40% | 30% | 25% | 10% | 4% | 1% |
|---|---|---|---|---|---|---|---|
| | PSNR (dB)/SSIM | | | | | | |
| SDA | 26.16/0.8048 | 24.97/0.7392 | 24.58/0.7127 | 24.58/0.7107 | 23.77/0.6489 | 21.75/0.5534 | 19.05/0.4522 |
| SDA-SDCS | **30.17/0.9026** | **29.90/0.8973** | **28.77/0.8704** | **28.13/0.8510** | **25.43/0.7338** | **23.38/0.6145** | **21.08/0.4865** |
| ReconNet | 30.85/0.8949 | 29.47/0.8647 | 27.95/0.8190 | 27.20/0.7914 | 23.98/0.6472 | 21.69/0.5557 | 18.96/0.4531 |
| ReconNet-SDCS | **33.27/0.9448** | **32.52/0.9355** | **31.04/0.9107** | **30.13/0.8921** | **26.46/0.7753** | **23.99/0.6502** | **21.20/0.5063** |
| CSNet$^+$ | **35.89/0.9677** | **33.96/0.9513** | **31.94/0.9251** | **30.91/0.9067** | **27.01/0.7949** | **24.41/0.6747** | 21.42/0.5261 |
| CSNet$^+$-SDCS | 34.91/0.9588 | 33.59/0.9462 | 31.80/0.9221 | 30.82/0.9043 | 26.97/0.7906 | 24.21/0.6692 | **21.48/0.5288** |
| ISTA-Net$^+$ | **34.92**/0.9510 | 32.87/0.9264 | 30.77/0.8901 | 29.64/0.8638 | 25.11/0.7124 | 21.82/0.5661 | 18.92/0.4529 |
| ISTA-Net$^+$-SDCS | 34.85/**0.9622** | **33.26/0.9465** | **31.38/0.9199** | **30.36/0.9003** | **26.56/0.7811** | **24.00/0.6555** | **21.24/0.5096** |
| DPDNN | 33.56/0.9373 | 32.05/0.9164 | 29.98/0.8759 | 28.87/0.8491 | 24.37/0.6863 | 21.80/0.5716 | 18.97/0.4544 |
| DPDNN-SDCS | **36.84/0.9708** | **34.91/0.9560** | **32.85/0.9323** | **31.74/0.9150** | **27.58/0.8069** | **24.78/0.6858** | **21.72/0.5319** |
| AMP-Net | **37.48/0.9744** | **35.34/0.9594** | **33.17/0.9358** | 32.01/**0.9188** | 27.82/0.8133 | 24.95/0.6949 | **21.90/0.5501** |
| AMP-Net-SDCS | 37.04/0.9720 | 35.18/0.9580 | 33.14/0.9354 | **32.04**/0.9187 | **27.84/0.8136** | **25.03/0.6967** | 21.87/0.5493 |

From Table 1 and Table 2, it can be found that compared with models without trained sampling matrices, although *model*-SDCS has only one model for reconstruction, it obtains better performance in terms of PSNR and SSIM at most test CS ratios. And compared with models that also apply trained sampling matrices (CSNet$^+$ and AMP-Net), *model*-SDCS can still obtain competitive performance at all test CS ratios with only a single model. Such a result implies the great potential of deep learning techniques and the sampling matrix training strategy. Therefore, we conclude that *model*-SDCS can effectively achieve SSR without changing the structure of the model.

Furthermore, it can be noticed from Table 1 and Table 2 that the above models combined with SDCS can have a good performance of SSR, which verifies the universality of SDCS. It is worth emphasizing that the purpose of the universality of SDCS is not to combine SDCS with all existing models, but to bring reconstruction models the effective SSR performance. This means that researchers can design reconstruction models at a single CS ratio. To achieve SSR, they only need to combine these models with SDCS.

In addition, Table 1 and Table 2 also verify the benefits of the sampling matrix after scalable training, which can be summarized into two points: 1) The trained sampling matrix can improve the performance of the reconstruction model, comparing models with SDCS and without SDCS. 2) The trained sampling matrix can adapt the reconstructed model to different CS ratios.

### 4.3 Comparison with SSR methods

In this subsection, we compare SDCS with two SSR methods: SCSNet (Shi et al., 2019b) and RACS (Lohit et al., 2018b).

Table 3: Parameter number of the reconstruction model of seven models.

| Parameter | SDA-SDCS | ReconNet-SDCS | CSNet$^+$-SDCS | ISTA-Net$^+$-SDCS | DPDNN-SDCS | AMP-Net-SDCS | SCSNet |
|---|---|---|---|---|---|---|---|
| Number | 6534 | 22914 | 370560 | 336978 | 1363712 | 229254 | 1110823 |

First, SCSNet is compared with SDA-SDCS, ReconNet-SDCS, CSNet$^+$-SDCS, ISTA-Net$^+$-SDCS, DPDNN-SDCS and AMP-Net-SDCS. Table 3 shows the parameter number of the seven models. Fig. 3 plots the average PSNR and SSIM of the seven models tested on the test set of BSDS500 at CS ratios from 1% to 50%. It can be noticed that except for DPDNN-SDCS, other models have fewer parameters than SCSNet and achieve SSR. And DPDNN-SDCS and AMP-Net-SDCS even outperform SCSNet, which shows the great potential of SDCS. Furthermore, deep unfolding models have better SSR performance than traditional deep learning models. For examples, AMP-Net-SDCS and DPDNN-SDCS outperform SDA-SDCS, ReconNet-SDCS and CSNet$^+$-SDCS, and ISTA-Net$^+$-SDCS outperform SDA-SDCS and ReconNet-SDCS. we conclude that deep unfolding models are more suitable for SSR to a certain degree due to the important role of the sampling matrix in the image reconstruction process.

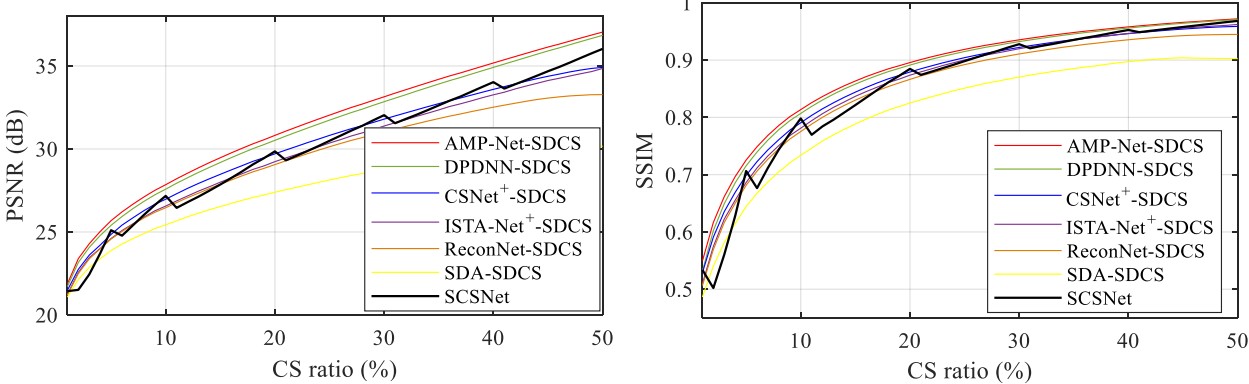

Figure 3: Comparison between SCSNet and six models with SDCS at different CS ratios.

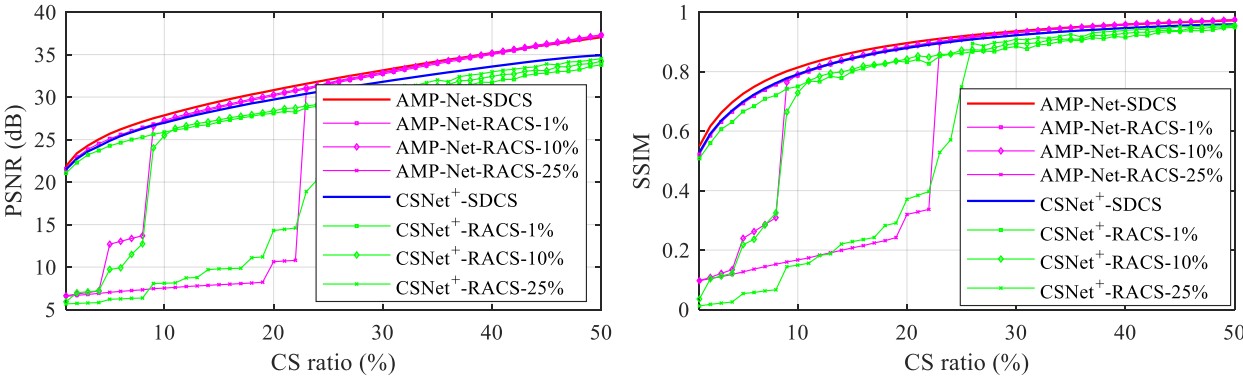

Figure 4: Comparison between SDCS and RACS at different CS ratios.

Second, SDCS is compared with RACS. Since with SDCS, AMP-Net outperforms other deep unfolding models and CSNet$^+$ outperforms other traditional deep learning models, we use AMP-Net and CSNet$^+$ as examples

to compare SDCS and RACS. In this subsection, the values of $R_K$ of RACS mentioned in 3.2 are 1%, 10% and 25%. Fig. 4 plots average PSNR and SSIM of AMP-Net-SDCS, CSNet$^+$-SDCS, AMP-Net-RACS-$R_K$ and CSNet$^+$-RACS-$R_K$ on the test set of BSDS500 at CS ratios from 1% to 50%, where $model$-RACS-$R_K$ denotes the model combined with RACS with the hyperparameter $R_K$. It can be noticed that when the CS ratio is lower than $R_K$, $model$-RACS-$R_K$ has bad performance. For AMP-Net, AMP-Net-SDCS outperforms all compared AMP-Net-RACS-$R_K$s when the CS ratio is lower than 30%. And for CSNet$^+$, CSNet$^+$-SDCS has better performance than all compared CSNet$^+$-RACS-$R_K$s at all CS ratios. Such a result implies that SDCS can generate a more appropriate combination of sampling matrix and model than RACS.

Fig. 5 shows the $Parrots$ images in Set11 reconstructed by different SSR models at different CS ratios. Fig. 5 is quite revealing in several ways. 1) AMP-Net-SDCS generates better results than SCSNet while maintaining fewer parameters which shows the great potential of SDCS. 2) $Model$-RACS-$R_K$ can not inherit the characteristics of original models well. For example, AMP-Net and CSNet$^+$ both have trainable deblocking operations, but AMP-Net-RACS-1% and CSNet$^+$-RACS-1% generates images with obvious blocking artifacts at CS ratios of 1%, 4% and 10%. However, AMP-Net-SDCS and CSNet$^+$-SDCS generate smooth images without blocking artifacts. Therefore, we conclude that models with SDCS can get good SSR performance. In particular, they can inherit the characteristics of original models.

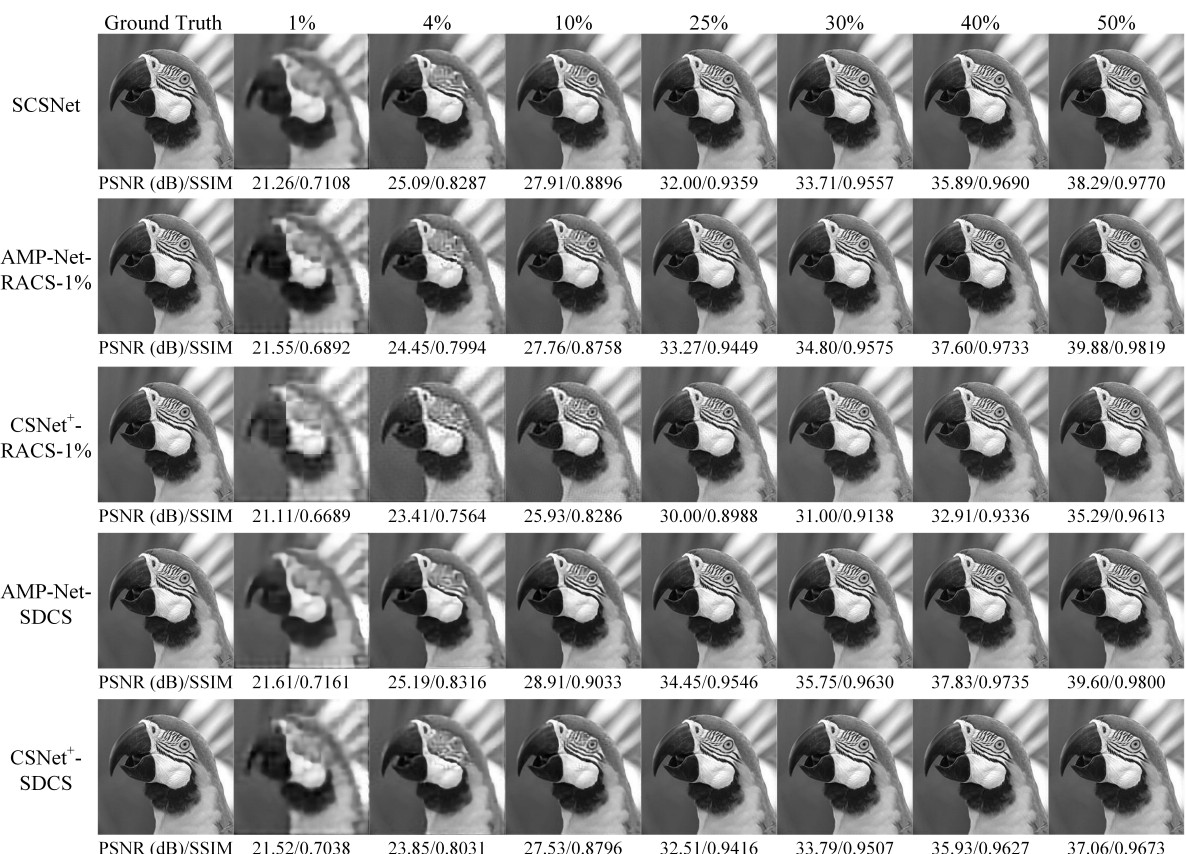

Figure 5: The $Parrots$ images in Set11 Reconstructed by different SSR methods at different CS ratios.

To further prove the effectiveness of SDCS, we compare AMP-Net-SDCS and CSNet$^+$-SDCS with AMP-Net and CSNet$^+$ which train their sampling matrices for one single CS ratio and apply the greedy algorithm in SCSNet (Shi et al., 2019b). In this subsection, the sampling matrices of AMP-Net and CSNet$^+$ are trained for the CS ratio of 50%. And their rows are rearranged using the greedy algorithm in SCSNet (Shi et al., 2019b) for better SSR. Fig. 6 plots the average PSNR and SSIM of four models tested on the test set of BSDS500 at CS ratios from 1% to 50%. It can be noticed that at the CS ratio of 50%, the specially trained

Table 4: The results of different models on the test set of BSDS500 with different SNRs.

| SNR | Method | 30% | 10% | 4% |
|---|---|---|---|---|
| | | PSNR (dB)/SSIM | | |
| 40dB | CSNet[+] | 31.89/0.9212 | 27.00/0.7942 | 24.51/0.6767 |
| | CSNet[+]-SDCS | 31.76/0.9219 | 27.03/0.7953 | 24.41/0.6768 |
| | AMP-Net | 32.93/0.9314 | 27.73/0.8088 | 24.94/0.6930 |
| | AMP-Net-SDCS | 32.83/0.9301 | 27.71/0.8083 | 24.95/0.6916 |
| 30dB | CSNet[+] | 31.58/0.9145 | 26.93/0.7863 | 24.43/0.6714 |
| | CSNet[+]-SDCS | 31.62/0.9178 | 26.95/0.7902 | 24.30/0.6720 |
| | AMP-Net | 31.40/0.9002 | 26.97/0.7761 | 24.46/0.6689 |
| | AMP-Net-SDCS | 31.61/0.9054 | 27.12/0.7818 | 24.57/0.6701 |
| 25dB | CSNet[+] | 30.87/0.8963 | 26.46/0.7665 | 24.12/0.6550 |
| | CSNet[+]-SDCS | 30.92/0.9023 | 26.66/0.7775 | 24.03/0.6624 |
| | AMP-Net | 29.83/0.8588 | 26.07/0.7328 | 23.82/0.6350 |
| | AMP-Net-SDCS | 29.90/0.8599 | 26.15/0.7349 | 23.78/0.6254 |
| 15dB | CSNet[+] | 26.84/0.7618 | 23.85/0.6337 | 21.69/0.5280 |
| | CSNet[+]-SDCS | 26.49/0.7492 | 24.15/0.6434 | 22.04/0.5479 |
| | AMP-Net | 26.43/0.8007 | 23.21/0.6003 | 21.66/0.5324 |
| | AMP-Net-SDCS | 25.29/0.6936 | 22.64/0.5486 | 20.52/0.4433 |

models can obtain better results than the models with SDCS, but such models have a bad performance at other CS ratios. However, models combined with SDCS perform well at all CS ratios. Therefore, we conclude that *model*-SDCS outperforms the model with a trained sampling matrix for a single CS ratio, and SDCS provides a more obvious improvement than the greedy algorithm of SCSNet under the condition of only one reconstruction model.

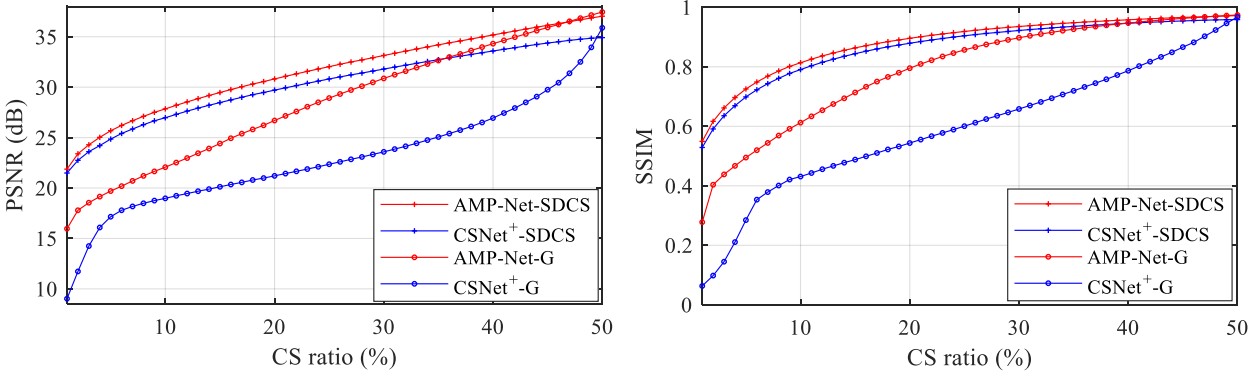

Figure 6: Comparison between two models with and without SDCS at different CS ratios. *model*-G is the model combined with the greedy algorithm in (Shi et al., 2019b).

## 4.4 Simulating the actual imaging conditions

In some practical conditions, noises may be introduced to the measurement **y**. To this end, we validate the anti-noise performance of SDCS in this subsection to simulate the actual CS imaging conditions. In detail, additive Gaussian white noises (Lepskii, 1991) are added to **y** of all datasets to train and test models in the subsection. And the signal-to-noise ratios (SNRs) are 40dB, 30dB, 25dB and 15dB. All results are obtained by testing 5 times on the test set and averaging.

Since with SDCS, AMP-Net outperforms the other deep unfolding models, and CSNet[+] outperforms other traditional deep learning models, we use AMP-Net and CSNet[+] as examples to validate the anti-noise performance of SDCS. Table 4 shows the average PSNR and SSIM by different models on the test set of BSDS500 with different SNRs at different CS ratios of 30%, 10%, and 4%. It can be noticed that in most cases, the original model and the model with SDCS have similar performance, which demonstrates that

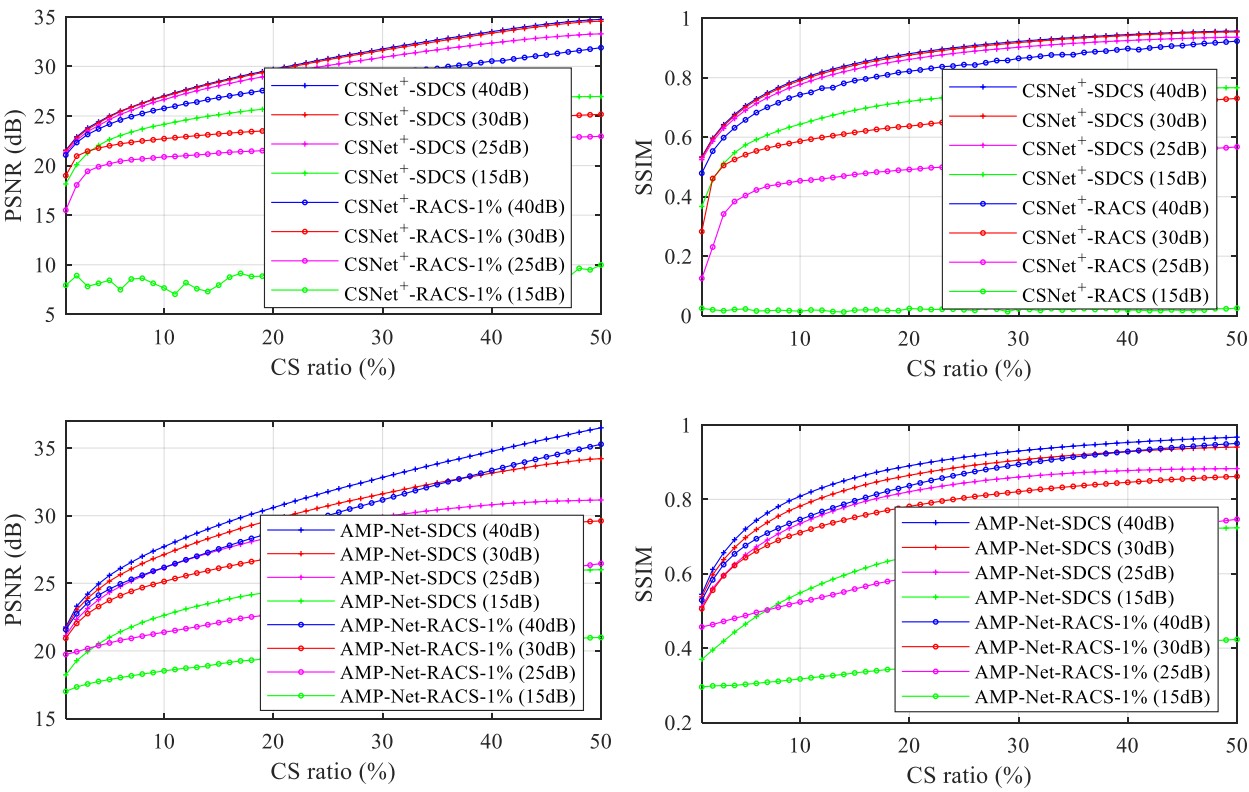

Figure 7: Comparison between SDCS and RACS with different SNRs.

SDCS will not weaken the anti-noise ability of the original model to a certain extent. The only exception is that AMP-Net outperforms AMP-Net-SDCS with a low SNR of 15dB, which means that the anti-noise performance of *model* may decline with a low SNR when it is combined with SDCS. Furthermore, combined with SDCS, a single model can be used to achieve sampling and reconstruction at multiple CS ratios, which further illustrates the advantage of SDCS.

Since *model*-RACS-1% has the most similar performance to *model*-SDCS in subsection 4.3, we compare *model*-RACS-1% with *model*-SDCS to further validate the anti-noise performance of SDCS. Fig. 7 shows the average PSNR and SSIM of CSNet$^+$-SDCS, CSNet$^+$-RACS-1%, AMP-Net-SDCS, AMP-Net-RACS-1% with different SNRs at CS ratios from 1% to 50%. In Fig. 4, when there is no noise, the maximum difference of the PSNR and the SSIM of *mdoel*-SDCS and *model*-RACS-1% are 1dB and 0.04, respectively. It can be noticed from Fig. 7 that as the SNR decreases, the performance of each model decreases, and the performance difference between *mdoel*-SDCS and *model*-RACS-1% also increases. In particular, the performance of CSNet$^+$-SDCS with an SNR of 15dB is even better than *model*-RACS-1% with an SNR of 30dB, and for CSNet$^+$-RACS-1%, the PSNR is even lower than 10dB and the SSIM is lower than 0.1, which may be due to that the traditional deep model combined with RACS cannot be well adapted to the condition of low SNR. Therefore, it can be concluded that SDCS has better anti-noise performance than RACS and is more suitable for imaging under actual conditions.

## 5   Conclusion

In this paper, for the visual image CS problem, we propose a general framework named SDCS to achieve SSR of deep-learning-based models. Besides of the initialization matrix and two sampling matrix masks, SDCS does not change the structure of the model. The proposed scalable training can generate an appropriate combination of the sampling matrix and the reconstruction model for efficient SSR. Experimental results show

that SDCS outperforms other SSR methods. Specifically, models with SDCS can inherit the characteristics of the original models, e.g. the deblocking ability. In addition, it is shown that SDCS can work well with additive noises.

However, SDCS has one shortcoming: $R_i$ being sampled uniformly during training makes the different training times of rows of the sampling matrix, which may affect the performance of SDCS. In the future, we will try to find a better way to generate $R_i$ and try some bigger datasets like ImageNet (He et al., 2016) to improve the power of SDCS. Furthermore, we will extend SDCS to high-dimensional CS problems which demand SSR. For example, snapshot compressive imaging (SCI) (Liu et al., 2018; Ma et al., 2019) is promising to use a single model to reconstruct hyperspectral images in different frequency bands, and some applications like transient imaging (Sun et al., 2018) and magnetic resonance imaging (MRI) (Liu et al., 2017; 2020) can obtain images at different ratios using one model with a binary sampling matrix. As different CS applications have different sampling and reconstruction strategies, which makes the current SDCS has to be updated to adapt to them.

## 6 Acknowledgements

This research is supported by National Natural Science Foundation of China (NSFC, No. 62171088, U19A2052, 62020106011), Medico-Engineering Cooperation Funds from University of Electronic Science and Technology of China (No. ZYGX2021YGLH215, ZYGX2022YGRH005).

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
