# OpenReview forum: "Scalable Deep Compressive Sensing"
_TMLR — Accepted by TMLR_

### Review · Reviewer_S97u · 2022-12-06

**Summary Of Contributions:**

The paper introduces a new training scheme to support compressed sensing with different compression ratios. The method learns a sampling matrix, a reconstruction initialization matrix, and the network parameters together. Authors propose a masking scheme to ensure that the same sampling matrix and reconstruction initiation matrix easily work across different compression ratios. Authors perform extensively experimentation with 6 different network architectures for reconstruction, and shows that when combined with the proposed method, these architectures outperform their counterpart trained with a vanilla scheme achieving state-of-the-art performance across multiple compression ratios. Furthermore, they show that training with the proposed scheme provides some robustness to addition to Gaussian noise.

**Audience:**

Yes

**Broader Impact Concerns:**

None.

**Claims And Evidence:**

Yes

**Requested Changes:**

See weakness above.

Minor nitpik: it might make sense to combine table 3 with tables 1 and 2 so that it is easier to read the results.

**Strengths And Weaknesses:**

The paper is well written, easy to read, and most claims are supported well with empirical evidence.

The idea is simple, and straightforward, and I appreciate the authors providing extensive empirical evidence that the idea works across different models and dataset. However, I've a few questions/comments about the set-up:

1. The idea of training A and B together with the network probably exists in the literature already. I might have missed this while the reading the paper, but can the authors provide more context around related work and confirm which elements (a) A matrix, (b) B matrix, (c) and the masking strategy is introduced in this paper vs what exists in the literature?
2. The results in Table 1 and 2 are very impressive, but it seems peculiar that a model optimized for a single compression ratio does not even come close to the performance of a model trained for multiple compression ratios. Can authors provide some insight on why this maybe the case? Are all experimental parameters the same (was same amount of effort spent on hyperparameter tuning)?
3. From algorithm 1, it seems like R_i s are sampled uniformly. Does this sampling strategy affect performance? For example, I might imagine that training with smaller R_i s in the initial epochs and expanding R_is as epoch increases will help get better performance for the top rows of A_i.
4. In Section 4.4 is Gaussian a realistic noise model? If yes, can authors kindly provide references and place it in context? Further, I assume that the models are not trained in the presence of Gaussian noise - but only evaluated - can the authors confirm this/make this clear in the paper?

---

> ### Author Response · Authors · 2023-02-13
> **Response to Reviewer S97u**
>
> # General Comments
> Response: We appreciate the positive feedback from the reviewer. Moreover, thank you for your comments and valuable suggestions. We have revised the paper accordingly and would like to address your concerns in the following.
> # Comment 1
> Response: Thanks for the comment. Yes, it is correct that The idea of training $A$ and $B$ together with the network already exist. In our paper, we intorduce a general scalable sampling and reconstruction(SSR) framwork named SDCS which can be applied on all end-to-end-trained models. In related works, we discussed some deep learning models, and concluded some methods train their sampling matrices as follows, and the content can be found in Section 3.1. In our paper, we concluded which methods train their sampling matrices as follows: ”Some of the above methods discuss the sampling matrix training strategies, including in traditional deep learning models ^[1], ^[2] or in the deep unfolding model ^[3], and they all train their initialization matrices.” Further more, we compare SDCS with other 2 typical SSR methods which apply other masking strategies in 3.2 as follows:
> 1) SCSNet: However, SCSNet has one weakness. Based on SCSNet, the existing deep learning models have to change their structure to achieve scalable reconstruction which would bring more burden to the hardware. However, SDCS needs only one model to achieve SSR and it can be applied to all end-to-end-trained models without changing their structures.
> 2) CRA: Compared with SDCS, CRA do not train the sampling matrix, and two reconstruction models would introduce more parameters. Furthermore, we emphasize that CRA is essentially a pluggable method, which can be combined with other SSR methods by applying a non-linear model for initialization and measurement completion. Therefore, the experimental comparison between SDCS and CRA is not the focus of our paper.
>
> # Comment 2
> Response: Thanks for the comment. This commet can be splitted to 2 subcomments:
> 1) Some models do not train their sampling matrices in their original papers, but models with SDCS train their sampling matrices, which brings stronger ability to obtain better samples for reconstruction. Compare with some models which also train their sampling matrices in the original papers, models with SDCS can still obtain competitive performance.
> 2) Yes, to ensure the objectivity and fairness of the experiment, for the same reconstruction model, all experimental parameters the same. In our experiments, different reconstruction models are combined with SDCS and RACS, and these models are trained on the conditions in their original papers. For example, For AMP-Net, the epoch size is 30, batch size is 32 in the original paper [3], so as for AMP-Net-SDCS and AMP-Net-RACS, the epoch size and batch size are 30 and 32 too.
> # Comment 3
> Response: Thank you for this comment! Yes, our method does have the above shortcoming. But such shortcoming do not cover up the advantages of SDCS as we claimed in the paper. However, your provide a very interesting insight, and improve the ability of SDCS possibly, it can be the next move of SDCS which is mentioned in the conclusion part as follows:
> ”However, SDCS has one shortcoming: $R_i$ beingsampled uniformly during training makes the different training times of rows of the sampling matrix, which may affect the performance of SDCS. In the future, we will try to find a better way to generate $R_i$ and try some bigger datasets like ImageNet ^[4] to improve the power of SDCS.”
> # Comment 4
> Response: Thanks for the comment. We are sorry that we do not describe it clearly. Yes, Gaussian is a realistic noise model, and we provide one reference. Furthermore, we both train and test the models in the presence of Gaussian noise, and related content can be found in Section 4.4 and is as follows: ”In detail, additive Gaussian white noises are added to $y$ in all datasets to train and test models in the subsection”.
>
> # Reference
>
> [1]: A. Mousavi, A. B. Patel, and R. G. Baraniuk, “A deep learning approach to structured signal recovery,” in The 53rd Annual Allerton Conference on Communication, Control, and Computing, pp. 1336–1343, IEEE, 2015.\
> [2]: W. Shi, F. Jiang, S. Liu, and D. Zhao, “Image compressed sensing using convolutional neural network,” IEEE Transactions on Image Processing, vol. 29, pp. 375–388, 2019.\
> [3]:  Z. Zhang, Y. Liu, J. Liu, F. Wen, and C. Zhu, “Amp-net: Denoising-based deep unfolding for compressive image sensing,” IEEE Transactions on Image Processing, vol. 30, pp. 1487–1500, 2021.\
> [4]:  K. He, X. Zhang, S. Ren, and J. Sun, “Deep residual learning for image recognition,” in The IEEE Conference on Computer Vision and Pattern Recognition, pp. 770–778, 2016.

---

### Review · Reviewer_s8cQ · 2022-12-31

**Summary Of Contributions:**

This paper proposes a technique that adapts deep networks for compressive sensing to multiple subsampling ratios. The proposed method involves learning common sampling and reconstruction matrices and using the corresponding rows/columns of the matrices for multiple subsampling ratios. This approach is modular and can be combined with any deep network architecture that performs compressive sensing and reconstruction of images/image patches. The authors perform experiments using different deep network architectures to demonstrate the efficacy at individual subsampling ratios. The authors also compare their methods to other proposals for robust and scalable compressive sensing and show their techniques are more parameter/compute efficient as well as achieving superior PSNR.

**Audience:**

Yes

**Broader Impact Concerns:**

None beyond the usual caveats and concerns that accompany models that are learned from data. The paper could perhaps emphasize that it does not address bias and fairness issues that could arise from using datasets that are not representative of certain populations.

**Claims And Evidence:**

Yes

**Requested Changes:**

1. Ablation experiment on learning $\mathbf{A,B}$ vs just training at different subsampling ratios without learning $\mathbf{A,B}$
2. Normalizing the amount of training data seen by SDCS vs baselines. If this is already the case, it should be mentioned/emphasized in the paper.

**Strengths And Weaknesses:**

Strengths:
1. SDCS seems to show results across multiple subsampling ratios that are superior to/as effective as the original techniques tuned to each subsampling ratio.
2. SDCS is modular and can be combined with any network architecture.

Weaknesses:
1. While the experiments demonstrate the efficacy of SDCS, the authors do not indicate whether this improvement can be attributed to the fact that SDCS trained networks simply see a lot more data ($B$ times the data that a single network sees). Especially at lower subsampling ratios, the number of samples that are used to train the models and sensing matrices could be a lot more than just $B$ times the data seen by a single network since higher subsampling ratios also influence the first few rows/columns of the sensing/reconstruction matrices. A clarification of this point, or an experiment that allows baselines to see more data will be helpful in pointing out the efficacy of the proposed method.

2. The authors do not perform any ablation studies that demonstrate the efficacy of different components. It would be interesting to know if the improvements are primarily due to learning the sampling/reconstruction matrices or due to the fact that the single network is trained on all the data. Implementing a version of SDCS that does not update the $\mathbf{A,B}$ matrices and only updates the parameter matrices $\Theta$ will answer this question.

3. It is unclear whether this reconstruction technique can generalize to sampling matrices that are not learned by the model. There may be scenarios where the measurements are generated by a randomized sampling technique and one may want to reconstruct the image from these measurements without access to the exact measurement matrix. How does SDCS perform in this scenario?

Presentation Improvements:
1. The paper could be written in a more accessible manner so that it can reach a wider audience than just the research community in deep learning for compressive sensing. A short primer explaining the scaling problem with deep networks for compressive sensing would help motivate the problem for the reader who is unfamiliar with the problems in this sub-field. In fact moving a version of Table 3 to the first few pages of the paper could give us a sense of the contributions of the paper before we dive into the algorithm.

2. Citations could use citep to put them in parantheses rather than just cite. There are a few places where the author names are repeated (Shi et al. Shi et al.; Lohit et al. Lohit et al.)

---

> ### Author Response · Authors · 2023-02-13
> **Response to Reviewer s8cQ**
>
> # Comment 1
> Response: Thanks for the comment.We have added to your proposed experiment. In Table 1, Models-SDCS* represents the experimental results that sampling matrix A and initialization matrix B do not participate in training. It can be seen from the experimental results that the reconstruction ability of the model A and B do not participate in the training is much reduced, so the effectiveness of our training strategy can be verified.
>
> # Comment 2
> Response: Thanks for the comment. Actually, we already normalize the amount of training data seen by SDCS vs baselines already. And we mentioned that in Section 4.1 as follows:
> ”In this paper CSNet+, AMP-Net, CSNet+-SDCS, AMP-Net-SDCS and SCSNet are trained on training set 1 due to the existence of trainable deblocking operations. SDA, ReconNet, ISTA − Net+, DPDNN, SDA-SDCS, ReconNet-SDCS, ISTA − Net+-SDCS and DPDNN-SDCS are trained on training set 2. And they are trained on the conditions in their original papers.”
>
>
> # Comment 3
> Response: Thank you for your comments. Table 3 shows the comparison between our work and the parameters of the two existing methods. We think it is unnecessary to put it in the front, and may lead the reader’s attention to the wrong place. Our figures 1 and 2 can help readers understand the relevant working principles and our work design.
> # Comment 4
> Response: Thank you for your comments! We listened to your suggestions and used \citep to change the author and reference conflicts.

---

> > ### Comment · Action_Editors · 2023-03-05
> > **Additional comment**
> >
> > After reviewing all of the reviewers' comments and authors' responses, I have the following additional two related questions:
> >
> > * Learning the sampling matrix $\bf A$ in conjunction with reconstruction methods has been thoroughly explored in the literature. Also, it is widely used in iterative algorithms (like the first step of ISTA with zero initialization) and deep learning-based methods to use the sampling matrix $\bf A$ to get an initialization $\bf A^\top \bf y$. Here the authors propose to learn a different matrix $\bf B$ for initialization $\bf B^\top \bf y$, but it is not very clear why this is desirable. The authors should conduct experiments to compare the performance of learning both $\mathbf{A}$ and $\mathbf{B}$ with learning only $\mathbf{A}$ and using $\mathbf{A}^\top \mathbf{y}$ as the initialization.
> >
> > * In many cases, AMP-Net appears to give the best performance, and the proposed strategy has no improvement for AMP. What distinguishes AMP-Net-SDCS from AMP-Net? Does AMP-Net also learn both $\mathbf{A}$ and $\mathbf{B}$? Overall, there are six models conducted in the experiments, but it is not very clear which of them use fixed sampling matrices/initialization and which ones already use learned ones, and how SDCS augments the six models. The authors could provide a summary of the settings for all six models to clarify this.

---

> > > ### Author Response · Authors · 2023-03-20
> > > **Response to Reviewer s8cQ**
> > >
> > > # Comment1:
> > > Response: Thank you for this comment. We are sorry that we do not explain clearly that we implement the $\bf B$ as the initialization matrix: We have consulted several deep-learning papers[1, 2, 3, 4], and it is shown that an independent initialization matrix is useful. For example, in [2], CSNet applies a linear convolutional layer to initialize the image, which also can be regarded as a trainable matrix. In [3], ISTA-Net obtains an initialization matrix form the training set and this matrix is different from $\bf A^\top$. In AMP-Net[4], a matrix $\mathbf{B}$ is applied to initialize the image. We add the explanation in Section 2.2, and the content is as follows:
> > >
> > > ” Based on some deep-learning-based models [2, 3, 4], an initialization matrix $\mathbf{B}$ is developed. ”
> > >
> > > # Comment2:
> > > Response: Thanks for this comment. we respond to this comment as the following three sub-comments.
> > >
> > > 1. What distinguishes AMP-Net-SDCS from AMP-Net?
> > >
> > > The difference between AMP-Net-SDCS and AMP-Net can be noticed in Fig.6 in the original. We compare AMP-Net-SDCS and CSNet+-SDCS with AMP-Net and CSNet+ which train their sampling matrices for one single CS ratio and apply the greedy algorithm in SCSNet[5]. The sampling matrices of AMP-Net and CSNet+ are trained for the CS ratio of 50%. And their rows are rearranged using the greedy algorithm in SCSNet [5] for better SSR. It can be noticed that AMP-Net-SDCS have much better performance than AMP-Net-G, and the such result shows the power of SDCS.
> > >
> > > 2. Does AMP-Net also learn both  $\mathbf{A}$ and  $\mathbf{B}$ ?
> > >
> > > Yes, AMP-Net also learns both $\mathbf{A}$ and  $\mathbf{B}$. And as we discussed in the paper, all the models are trained on the conditions in their original papers.
> > >
> > > 3. Overall, there are six models conducted in the experiments, but it is not very clear which of them use fixed sampling matrices/initialization and which ones already use learned ones, and how SDCS augments the six models. The authors could provide a summary of the settings for all six models to clarify this.
> > > We add some content to summarize, and it can be found in Sections 4.1 and 4.2. All content is as follows:
> > >
> > >    (1) SDA, ReconNet, ISTA − Net+ and DPDNN do not train the sampling matrix in their original matrix, and CSNet+ and AMP-Net have the trainable sampling matrix. Furthermore, the above six models have the same initialization matrix as equation (3) in this paper.
> > >
> > >    (2) The way to combine these models with SDCS is described in Algorithm1.
> > >
> > > [1] S. Lohit, K. Kulkarni, R. Kerviche, P. Turaga, and A. Ashok, “Convolutional neural networks for noniterative reconstruction of compressively sensed images,” IEEE Transactions on Computational Imaging, vol. 4, no. 3, pp. 326–340, 2018.、
> > >
> > > [2] W. Shi, F. Jiang, S. Liu, and D. Zhao, “Image compressed sensing using convolutional neural network,” IEEE Transactions on Image Processing, vol. 29, pp. 375–388, 2019.
> > >
> > > [3] J. Zhang and B. Ghanem, “ ISTA-Net : Interpretable optimization-inspired deep network for image compressive sensing,” in The IEEE Conference on Computer Vision and Pattern Recognition, pp. 1828–1837, 2018.
> > >
> > > [4] Z. Zhang, Y. Liu, J. Liu, F. Wen, and C. Zhu, “Amp-net: Denoising-based deep unfolding for compressive image sensing,” IEEE Transactions on Image Processing, vol. 30, pp. 1487–1500, 2021.
> > >
> > > [5] W. Shi, F. Jiang, S. Liu, and D. Zhao, “Scalable convolutional neural network for image compressed sensing,” in The IEEE Conference on Computer Vision and Pattern Recognition, pp. 12290–12299, 2019.

---

> > > > ### Comment · Action_Editors · 2023-03-21
> > > > **Thank you for the clarification**
> > > >
> > > > Thank you for the responses and for revising the paper. They answered my questions.

---

### Review · Reviewer_417w · 2023-01-30

**Summary Of Contributions:**

The paper proposes a framework for solving (with a single model) Compressed Sensing problems in which different subsampling ratios might be encountered. The paper considers the supervised setting for solving inverse problems, i.e. a model is trained to reconstruct an image from a corrupted observation. The empirical finding of prior work is that the best performance is obtained when different models are trained for (significantly) different subsampling rations. The current paper considers this a prior work limitation mainly because of hardware (storage?) and training constraints. The paper proposes to solve this problem by using a trainable sampling (forward) matrix, A. During training, the matrix is masked at different ratios to reconstruct images, with solution initializations arriving from masking (at different ratios) of an initialization matrix B.



**Audience:**

Yes

**Broader Impact Concerns:**

I do not have any ethical concerns.

**Claims And Evidence:**

Yes

**Requested Changes:**

Critical for acceptance:
* I would recommend citing, discussion and potential comparison to the unsupervised approach for solving inverse problems, i.e. the approach of using pre-trained generative networks as priors.
* Experimental section should be strengthened. I think it would be best to include some results on classic datasets, such as CIFAR-10 or ImageNet. I understand that the computational requirements might be an issue, but I think at least CIFAR-10 should be doable in the described hardware and it will be a convincing experiment that this framework scales.
* I think it would be valuable to the paper to evaluate also performance when the forward (sampling) matrix is given. This will also help getting a better sense of how this method performs in standard benchmarks and compared to recent state-of-the-art approaches for solving specific inverse problems. For example, a valuable comparison would be with MaxVIT (MAXIM).

Not so critical for acceptance:
* Fix several typos and improve the delivery of the paper. Examples:
     * "bring the model the ability" -> "bring to the model the ability".
     * in the related work section, there are repetitions in the citations, such as Lohit et. al Lohit et. al (2018b)
     * The first paragraph of 3.1. feels a bit disconnected with the paper since it just lists architectural innovations, which is orthogonal to the proposed framework.
* I think some parts could be better explained for the unfamiliar reader. E.g. why is it better to have one sampling matrix (that gets masked every time?).  Also, it would help define what the block-based visual image CS problem is.
* It would be nice to explain why the initial value for the problem to be solved is chosen to be linear matrix times the measurements. I could imagine "rough" non-linear reconstruction approximations that would later be improved.

**Strengths And Weaknesses:**

Strengths:
* The problem that the authors study is relevant.
* The paper builds upon a large line of prior work.
* The proposed solution is intuitive.
* The results show that the method outperforms both single model and previously proposed SSR solutions.


Weaknesses:
* The paper ignores a body of work based on generative networks for solving inverse problems. A different approach to the supervised learning route is to use a generative model as a prior for the domain and then find a solution from the network that explains the measurements. This approach has found a lot of empirical success (especially recently with the rise of powerful generative networks) and doesn't have the problem of different ratios which is the topic of investigation of the current paper. This line of work should be at least cited (if not compared to). For example, the Bora et. al. (2017) paper, is incorrectly cited by the authors as a supervised learning approach for solving inverse problems.
* The paper's presentation could be improved (see Requested Changes for a list of typos, suggested improvements).
* Experiments are limited. The biggest training set contains only 200 images. Also, since the testing on Set11 isn't it considered out-of-distribution testing since the training images were colorful? What is the point of the experiments on this test set? Is this a claim of better out-of-distribution performance?
* From the paper, I gathered that the forward matrix is free to be chosen at will (which seems to be one of the main innovations of the paper). It would be cool to see what the learned forward matrix is, in the sense of how the images look visually after the degradation (sampling) at different rations.
* Also, in many problems, the forward matrix is fixed, i.e. in physical systems. It would be cool to evaluate performance on this setting, e.g. maybe in the setting where A is a random inpainting matrix (with different probabilities of masking based on the corresponding subsampling ratio).
* One of the main arguments of the paper against having separate models based on the subsampling ratio is the storage of the trained models. But these models will not be doing something entirely different so it is possible that some method for efficient parameter tuning would be efficient. For example, the [LoRA](https://arxiv.org/abs/2106.09685) that was proposed for efficient tuning of LLMs, has been lately applied to Computer Vision and Stable Diffusion. It would be nice to hear the author's perspective on this.

---

> ### Author Response · Authors · 2023-02-13
> **Response to Reviewer 417w**
>
> # Comment 1
> Response: Thank you for your comments. This is really a cool idea. But we visualized the trained A,B and found that we couldn’t understand what had changed in it.We then subtract the untrained matrix from the trained matrix, as shown below. We can see that A,B does change, but we can’t understand how it changes. The relevant pictures are placed in the support material.
>
> # Comment 2
> Response: Thank you for your comments. Because our method is an end-toend training, it does not need to manually adjust parameters. And because the network is embedded in the optimization algorithm, the network parameters are relatively small. In addition, we have not investigated the method you said, please give us specific references.
>
>
> # Comment 3
> Response: Thank you for your comments.This kind of method is similar to the plug-and-play model. Although this kind of model can provide better experimental results, it is not the end-to-end training network, and this kind of model relies too much on the ability of the prior model. So we’re not comparing these kinds of methods.
> # Comment 4
> Response: Thank you for your comments. We understand your concern, but our datasets are sufficient for the experiments. Actually, BSDS500 and Set11 contain high-resolution images, and are classical datasets in the deep-learning image CS problem. Further more they are already applied in many good papers. [2, 3, 5]. Moreover, we generate two training datasets which contain 89600 images and 195200 images which have larger number than CIFAR-10. But we will try ImageNet in the future, and we mention that in the conclusion part as follows:
> ”In the future, we will try to find a better way to generate Ri and try some bigger datasets like ImageNet [4] to improve the power of SDCS.”
> # Comment 5
> Response: Thank you for your comments. We have cited the paper [6] and the method has strong generative modeling capabilities. However, the method in this paper is mainly applied to classification and target detection, but we mainly proposed a scalable image compressed sensing method, which can make the model more convenient to use at different sampling rates.
> # Comment 6
> Response: Thank you for your comments. The relevant changes and explanations are as follows:
> • ”bring the model the ability” -> ”bring to the model the ability”.
> • For the problem of repeated references, we replace \cite with \citep. Now, the author and the reference are separated by brackets, such as Lohit et al. (Lohit et al., 2018a)
> • Sections 3.1 and 3.2 describe two different learning modes. Section 3.1 describes several popular deep learning methods, and Section 3.2 describes several tasks for scalable sampling rate learning. Both of them have inspired our work at different levels, so subsection 3.1 is not disconnected from the full text.
> # Comment 7
> Response: Thank you for your comments. Our special design for sampling matrix can realize unified learning mode with different compression rates. We have added the relevant description in section 2.1, which can help readers understand the advantages of this design faster.
>
> # Comment 8
> Response: Thank you for your comments. At present, the mainstream research on compression sensing is to regard it as a linear inverse problem, aiming at restoring high-dimensional signals from a small number of linear measurements. Research on nonlinearity may be our future focus.
>
> # Reference
>
> [1]: A. Mousavi, A. B. Patel, and R. G. Baraniuk, “A deep learning approach to structured signal recovery,” in The 53rd Annual Allerton Conference on Communication, Control, and Computing, pp. 1336–1343, IEEE, 2015.\
> [2]: W. Shi, F. Jiang, S. Liu, and D. Zhao, “Image compressed sensing using convolutional neural network,” IEEE Transactions on Image Processing, vol. 29, pp. 375–388, 2019.\
> [3]:  Z. Zhang, Y. Liu, J. Liu, F. Wen, and C. Zhu, “Amp-net: Denoising-based deep unfolding for compressive image sensing,” IEEE Transactions on Image Processing, vol. 30, pp. 1487–1500, 2021.\
> [4]:  K. He, X. Zhang, S. Ren, and J. Sun, “Deep residual learning for image recognition,” in The IEEE Conference on Computer Vision and Pattern Recognition, pp. 770–778, 2016.\
> [5]: J. Zhang and B. Ghanem, “ ISTA-Net : Interpretable optimization-inspired deep network for image compressive sensing,” in The IEEE Conference on Computer Vision and Pattern Recognition, pp. 1828–1837, 2018.\
> [6]: Z. Tu, H. Talebi, H. Zhang, F. Yang, P. Milanfar, A. Bovik, and Y. Li, “Maxvit: Multi-axis vision transformer,” in Computer Vision–ECCV 2022: 17th European Conference, Tel Aviv, Israel, October 23–27, 2022, Proceedings, Part XXIV, pp. 459–479, Springer, 2022.

---

### Comment · Action_Editors · 2023-01-30
**Discussion phase has begun**

3 reviews have been submitted and the discussion phase has begun. Thanks a lot to all the reviewers for their efforts in reviewing this paper.

In this phase, authors can take into account the reviewers’ feedback, answer any questions, and update the manuscript accordingly.

The reviewers can continue the discussion with the authors to gather all the information needed for submitting a decision recommendation.

---

### Decision · Action_Editors · 2023-04-04

**Recommendation:** Accept as is

**Comment:**

The main contribution of this paper is the proposal of a flexible deep compressive sensing approach that can accommodate multiple subsampling ratios. Extensive experiments support the performance of the proposed approach.  the paper received mixed recommendations, but the majority is leaning toward acceptance. Together with the fact that better performance is offered by the proposed approach in many scenarios and the claims are supported by accurate and clear evidence, I recommend acceptance.

**Audience:**

This work would be of interest to researchers working on deep learning and image processing.

**Claims And Evidence:**

In this paper, the authors investigate the application of deep networks to compressive sensing across multiple subsampling ratios. Their proposed method involves learning shared sampling and reconstruction matrices, and utilizing the corresponding matrix elements for various subsampling ratios. This modular approach is adaptable to any deep network architecture capable of compressive sensing and reconstruction of images or image patches. The authors conducted experiments using various deep network architectures to evaluate the effectiveness of their approach at different subsampling ratios. Additionally, they compared their method to alternative proposals for robust and scalable compressive sensing, and demonstrated its superior efficiency in terms of parameters and computation, while achieving higher PSNR.